# Duplex ultrasound findings and clinical outcomes of carotid restenosis after carotid endarterectomy

Hyangkyoung Kim[1], Eunae Byun[1], Min-Jae Jeong[2], Hee Sun Hong[1], Youngjin Han[1], Tae-Won Kwon[1], Yong-Pil Cho[1] *

1 Department of Surgery, Asan Medical Center, University of Ulsan College of Medicine, Seoul, Republic of Korea, 2 Department of Surgery, GangNeung Asan Hospital, University of Ulsan College of Medicine, Gangneung, Republic of Korea

* ypcho@amc.seoul.kr

**Data Availability Statement:** All relevant data are within the manuscript and its Supporting Information files.

## Abstract

This study aimed to describe the duplex ultrasound (DUS) findings associated with carotid restenosis after carotid endarterectomy (CEA) and to determine whether carotid restenosis is associated with the clinical outcomes of CEA. Between January 2007 and December 2016, a total of 660 consecutive patients who underwent 717 CEAs were followed up at our hospital with DUS surveillance for at least 3 years after CEA. These patients were analyzed retrospectively for this study. Following CEA, restenosis was defined as the development of ≥50% stenosis, diagnosed on the basis of DUS findings of the luminal narrowing and velocity criteria. The study outcomes were defined as restenosis of the ipsilateral carotid artery after CEA and late (>30days) fatal or nonfatal stroke ipsilateral to the carotid restenosis. During the median follow-up period of 74 months, the restenosis incidence was 2.8% (20/717), and there were 2 strokes (2/20, 10%) ipsilateral to the restenosis after CEA; reintervention was performed for 11 patients with carotid restenosis (55%). Within 2 years after CEA, restenosis was identified in 9 cases (45%, 9/20), and 8 reinterventions (72.7%, 8/11) were performed. According to DUS findings, the morphologic characteristics of carotid restenosis were different from the preoperative plaque morphology. Among the 20 carotid restenosis cases, we observed the following DUS patterns: homogenous isoechoic restenosis (n = 14, 70%), homogenous hypoechoic (n = 2, 10%), isoechoic with hypoechoic surface (n = 3, 15%), and hypoechoic with isoechoic surface (n = 1, 5%). Although 9 carotid restenosis patients received prophylactic reintervention to mitigate the progression of restenosis, the 2 symptomatic restenosis patients had isoechoic lesions with hypoechoic surfaces on DUS. On Kaplan–Meier survival analyses, in terms of stroke-free survival rates, there was a higher risk of stroke among patients with carotid restenosis compared with patients without restenosis, with a non-significant trend (P = 0.051). In conclusion, most carotid restenoses were identified within 2 years after CEA, and there was a non-significant trend toward a higher risk of stroke among patients with carotid restenosis.

**Funding:** The author(s) received no specific funding for this work.

**Competing interests:** The authors have declared that no competing interests exist.

## Introduction

Despite the well-proven efficacy of carotid endarterectomy (CEA) for stroke prevention for both symptomatic and asymptomatic patients with moderate to severe carotid stenosis [1], controversy exists whether restenosis after CEA is associated with an increased risk of recurrent ipsilateral stroke [2]. In many reports, carotid restenosis after CEA has been usually defined as a diameter reduction >50%; however, the clinical relevance of this definition has not yet been established [3]. Although restenosis is generally benign and usually does not require reintervention [4], the incidence of symptomatic restenosis has been reported to range from 0.6% to 3.6% among carotid restenoses [5]. The Carotid Revascularization Endarterectomy versus Stenting Trial (CREST) reported that restenosis >70% was associated with a significantly higher prevalence of recurrent stroke after CEA [3]. Carotid restenosis can occur early (within 2 years after CEA) or late (>2 years after CEA) [6, 7]. The reported incidence of carotid restenosis is highest within 1 year after CEA, ranging from 6% to 36%, and then decreasing thereafter [8]; incidence varies depending on the precise definition of restenosis and the duration of follow-up in published series [9]. Moreover, although patients have been regularly followed up by duplex ultrasound (DUS) imaging after CEA, whether routine surveillance using DUS provides an opportunity for timely reintervention to reduce the risk of neurologic events among asymptomatic patients is debatable [10]. Accordingly, follow-up strategies and management indications for restenosis still remain unclear [5, 11].

This single-center study aimed to describe the DUS findings associated with carotid restenosis after CEA and to determine whether carotid restenosis is associated with the clinical outcomes of CEA.

## Materials and methods

### Study design and population

This single-center, observational study was conducted retrospectively using data extracted from the medical records of patients who had undergone CEA at our hospital. Approval for data collection and publication was granted by the institutional review board (IRB No. 2019–0820) of our hospital, which waived the requirement for written informed consent because of the retrospective nature of the study. All methods were performed in accordance with the relevant guidelines and regulations.

A total of 790 consecutive patients who underwent 938 CEAs, between January 2007 and December 2016 at our hospital, were screened for inclusion in this study. Patients were eligible for inclusion if they had undergone an initial CEA for symptomatic or asymptomatic significant carotid stenosis with DUS follow-up of at least 3 years after CEA at our hospital. The demographic characteristics, risk factors of interest, clinical characteristics and outcomes, and DUS findings for all consecutive patients were recorded in an Excel (Microsoft Corp., Redmond, WA, USA) database and analyzed retrospectively.

### Index procedure and DUS surveillance

All CEAs were performed according to the Vascular Surgery guidelines for management of extracranial carotid disease [12], with the diagnosis of significant carotid bifurcation stenosis (50%–99% luminal narrowing) as defined by the DUS criteria suggested by the North American Symptomatic Carotid Endarterectomy Trial [13] and the Society of Radiologists in Ultrasound Consensus Conference [14]. The surgical procedures were performed as previously detailed [15]. A conventional endarterectomy with patch angioplasty in the standard fashion under general anesthesia with routine carotid shunting or regional anesthesia with selective

shunting was performed. Postoperatively, all patients were given dual antiplatelet therapy with a statin in combination with stringent blood pressure control and close observation in an intensive care unit for at least 24 hours. All patients were followed up both clinically and by magnetic resonance imaging with angiography or carotid DUS before discharge.

Follow-up visits, with laboratory evaluations and carotid DUS (iU22, Philips Ultrasound, Bothell, WA, USA) as well as independent neurological examinations, were scheduled at 6 and 12 postoperative months, and annually thereafter. Once stability had been established for over 3 years, surveillance was performed at longer intervals of about 2 years.

### Study outcomes and definitions

DUS reports were recorded by dedicated, board-certified vascular surgeons, and all recorded images were independently re-evaluated—for carotid plaque characterization and morphologic characteristics of carotid restenosis—by 1 specialized vascular surgeon and 1 experienced radiology technologist. Risk factor variables were defined as previously described [16].

The study outcomes were defined as restenosis of the ipsilateral carotid artery after CEA and late (>30days) fatal or nonfatal stroke or transient ischemic attack ipsilateral to the carotid restenosis. Following CEA, restenosis was defined as the development of ≥50% stenosis, diagnosed on the basis of DUS findings of luminal narrowing and velocity criteria with a peak systolic velocity (PSV) threshold ≥125 cm/s or an internal carotid artery (ICA)/common carotid artery (CCA) PSV ratio >2.0 [14]. Neurologic events were defined as previously detailed [15, 16].

### Statistical analysis

Categorical variables are reported as frequencies or percentages, and continuous variables are reported as medians and interquartile ranges (IQRs). The Mann–Whitney $U$ test was used for comparisons of non-normally distributed continuous variables. Categorical variables were compared using the chi-square test or Fisher's exact test, as appropriate, whereas continuous variables were compared using Student's $t$-test. Study outcomes—restenosis-free and stroke-free survival rates—were analyzed using Kaplan–Meier curves. A Cox regression model with backward stepwise selection was constructed to identify clinical variables associated with restenosis, and hazard ratios with 95% confidence intervals (CIs) are reported. Variables with a P-value <0.1 on univariable analysis were included in the multivariable analysis. A P-value <0.05 was considered statistically significant. Statistical analyses were performed using SPSS Statistics for Windows, version 21.0 (IBM Corp., Armonk, NY, USA).

### Results and discussion

According to the inclusion criteria, 660 patients who underwent 717 CEAs and were followed up with DUS surveillance for at least 3 years after CEA at our hospital were consecutively enrolled in this study. The baseline and clinical characteristics of the study sample according to the occurrence of carotid restenosis after CEA are presented in **Table 1**. With regard to baseline demographics and atherosclerotic risk factors, patients with carotid restenosis were more likely to be female (27.8% versus 10.9%; P = 0.04) and have dyslipidemia (77.8% versus 53.0%; P = 0.053) than those without carotid restenosis. During the perioperative period (within 30 days after CEA), there were 6 minor strokes (6/717, 0.8%) and 4 major strokes (4/717, 0.6%) among the entire study sample; however, among patients with perioperative neurologic events, there were no late (>30 days after CEA) restenoses during follow-up. During the median follow-up period of 74 months (IQR, 53.5–99.5 months), the restenosis incidence was 2.8% (20/717), including 2 patients with bilateral restenoses after staged bilateral CEAs. There

**Table 1. Baseline and clinical characteristics of the study sample according to carotid restenosis after CEA.**

|  | Total | Restenosis | No restenosis | P value |
|---|---|---|---|---|
| Patients (n) | 660 | 18 (2.7) | 642 (97.3) |  |
| Age (year) | 68 (63–73) | 66 (55–70) | 69 (63–73) | 0.15 |
| Male sex | 585 (88.6) | 13 (72.2) | 572 (89.1) | 0.04 |
| BMI (kg/m$^2$) | 23.9 (22.1–25.8) | 24.4 (23.0–26.4) | 23.9 (22.0–25.8) | 0.23 |
| Atherosclerosis risk factors |  |  |  |  |
| Hypertension | 493 (74.7) | 15 (83.3) | 479 (74.6) | 0.76 |
| Diabetes mellitus | 262 (39.7) | 8 (44.4) | 255 (39.7) | 0.94 |
| Dyslipidemia | 354 (53.6) | 14 (77.8) | 340 (53.0) | 0.053 |
| Smoking | 370 (56.1) | 9 (50.0) | 361 (56.0) | 0.64 |
| Medical history |  |  |  |  |
| CAD | 152 (23.0) | 2 (11.1) | 150 (23.4) | 0.39 |
| PCI/CABG | 117 (17.7) | 1 (5.6) | 116 (18.1) | 0.22 |
| CKD | 122 (18.5) | 2 (11.1) | 120 (18.7) | 0.55 |
| PAOD | 53 (8.0) | 0 (0) | 53 (8.3) | 0.39 |
| Cancer | 34 (5.1) | 2 (11.1) | 32 (5.0) | 0.17 |
|  | Total | Restenosis | No restenosis | P value |
| CEA (n) | 717 | 20 (2.8) | 697 (97.2) |  |
| Degree of carotid stenosis[a] |  |  |  |  |
| 50–69% | 7 (1.0) | 0 | 7 (1.1) | 0.65 |
| 70–99% | 710 (99.0) | 20 (100) | 690 (99.0) |  |
| Symptomatic stenosis | 330 (46.0) | 10 (50.0) | 320 (45.9) | 0.82 |

Continuous data are presented as medians and interquartile ranges; categorical data are presented as numbers (%).

BMI, body mass index; CABG, coronary artery bypass graft; CAD, coronary artery disease; CEA, carotid endarterectomy; CKD, chronic kidney disease; PAOD, peripheral arterial occlusive disease; PCI, percutaneous coronary intervention.

[a] Preoperative degree of carotid stenosis.

were 12 late neurologic events ipsilateral to the CEA (12/697, 1.7%) among patients without carotid restenosis and 2 (2/20, 10%) among those with restenosis (P = 0.055).

The clinical and DUS characteristics of the patients with carotid restenosis after CEA are presented in **Table 2**. According to the time interval from CEA to restenosis, the restenosis was identified 20% (n = 4) of the time in the first year, 25% (n = 5) in the second year, and each one per year thereafter. Eleven carotid restenosis patients (11/20, 55%) received reintervention—all with carotid artery stenting (CAS)—due to neurologic events (2 restenoses) and progression of restenosis (9 restenoses), and among them, 8 reinterventions were performed within 2 years after CEA. There were no significant differences in degree of restenosis (median, 80% [IQR, 70–80%] versus median, 65% [IQR, 52–85%]; P = 0.20) and PSV (median, 344.0 cm/s [IQR, 157.5–466.0 cm/s] versus median, 442.5 cm/s [IQR, 165.0–548.0 cm/s]; P = 0.52) between restenosis with and without reintervention. However, the time interval from CEA to restenosis was significantly longer in association with restenosis without reintervention (median, 77 months [IQR, 43.4–122.2 months]) than with reintervention (median, 16 months [IQR, 5.7–68.8 months]) (P = 0.04). During the median follow-up duration of 18 months (IQR, 6–48 months) after reintervention, there were no recurrent restenoses after reintervention.

According to DUS findings, lesion sites and morphologic characteristics of carotid restenosis were different from the preoperative findings. Among the 20 carotid restenoses, most stenotic lesions were proximal ICA in 80% (n = 16), CCA in 5% (n = 1), and carotid bulb in 15% (n = 3) on preoperative DUS, whereas there were 85% (n = 17) in the ICA distal to the operated-on

**Table 2. Clinical and DUS characteristics of the patients with carotid restenosis after CEA.**

|  | Characteristics |
|---|---|
| Degree of carotid restenosis |  |
| 50–69% | 6 (30.0) |
| 70–99% | 12 (60.0) |
| Occlusion | 2 (10.0) |
| Velocity |  |
| ICA PSV (cm/s) | 400.5 (161.8–544.0) |
| ICA/CCA PSV ration | 4.85 (2.38–7.00) |
| Location of restenosis |  |
| ICA distal to CEA | 17 (85.0) |
| Distal CCA | 3 (15.0) |
| Morphologic characteristics |  |
| Homogenous isoechoic restenosis | 14 (70.0) |
| Homogenous hypoechoic restenosis | 2 (10.0) |
| Isoechoic with hypoechoic surface | 3 (15.0) |
| Hypoechoic with isoechoic surface | 1 (5.0) |
| Indications for reintervention |  |
| Recurrent TIA | 1 (9.1) |
| Stroke | 1 (9.1) |
| Progression of restenosis | 9 (81.8) |

Continuous data are presented as medians and interquartile ranges; categorical data are presented as numbers (%).
CCA, common carotid artery; CEA, carotid endarterectomy; DUS, duplex ultrasound; ICA, internal carotid artery; TIA, transient ischemic attack; PSV, peak systolic velocity.

carotid artery and 15% (n = 3) in the distal CCA on follow-up DUS. In terms of plaque morphology, there were echogenic (n = 9, 45%), isoechoic (n = 7, 35%), and echolucent (n = 4, 20%) plaques on preoperative DUS, whereas there were homogenous isoechoic restenoses (n = 14, 70%), homogenous hypoechoic restenoses (n = 2, 10%), isoechoic restenoses with hypoechoic surfaces (n = 3, 15%), and hypoechoic restenosis with isoechoic surface (n = 1, 5%) on follow-up DUS (**Fig 1**). Although 9 carotid restenosis patients received prophylactic reintervention to mitigate the progression of restenosis, the 2 symptomatic restenosis patients had isoechoic lesions with hypoechoic surfaces on DUS. On follow-up DUS, there were changes in restenosis thickness over time but no changes in shape and echogenicity in any of the restenoses.

Univariable analyses identified female sex as significantly associated with carotid restenosis (P = 0.03). Cox regression analysis revealed female sex as associated with a 3.05-fold increased likelihood of restenosis after CEA (95% CI, 1.09–8.47; P = 0.03) (**Table 3**). The low number of late neurologic events ipsilateral to the restenoses (n = 2) limited our ability to assess the correlation between clinical variables and late neurologic events with a Cox regression model.

On Kaplan–Meier survival analyses of the cumulative event-free rates of the entire study sample, the estimated mean restenosis-free survival duration was 153.7 months (95% CI, 151.3–156.1 months), and the estimated freedom from restenosis rate at 10 years was 98.7% (**Fig 2A**). The estimated mean stroke-free survival duration was 155.2 months (95% CI, 153.3–157.2 months), and the estimated rate of freedom from any type of ipsilateral stroke at 10 years was 95.9% (**Fig 2B**). In terms of stroke-free survival rates, there was a higher risk of stroke among patients with carotid restenosis compared with patients without restenosis, with a non-significant trend (P = 0.051) (**Fig 2C**).

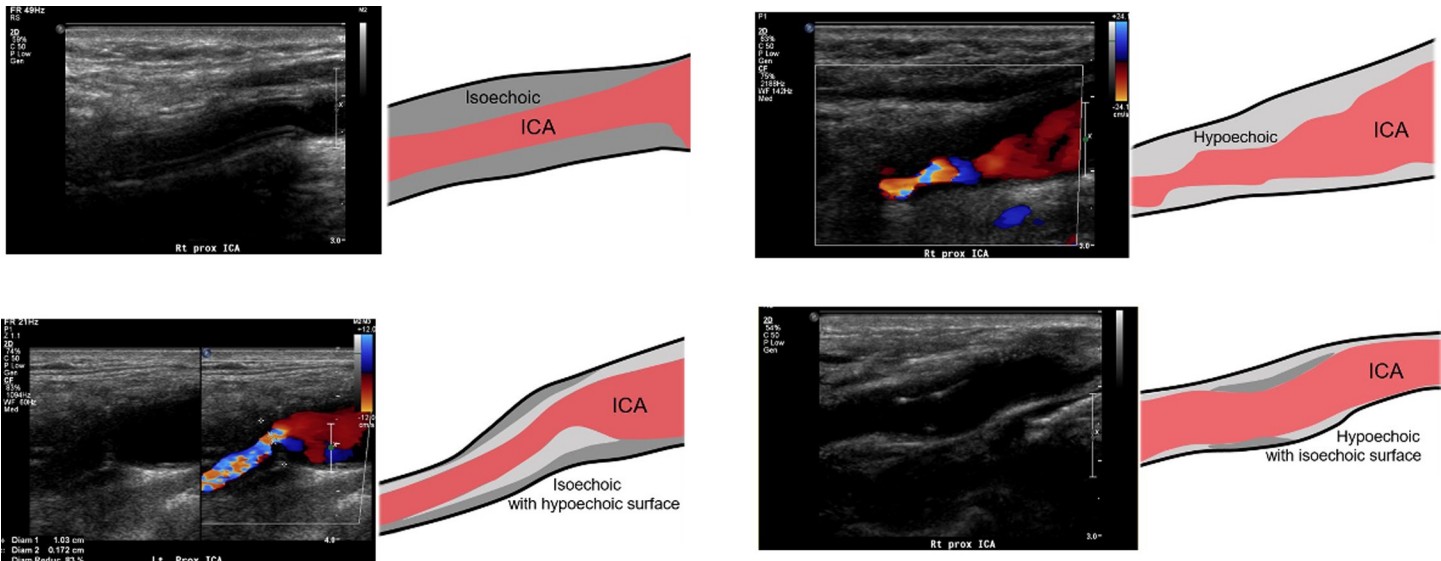

**Fig 1. Schematic representative figures of morphologic characteristics of carotid restenosis on DUS. (A)** Homogenous isoechoic restenosis, **(B)** homogenous hypoechoic, **(C)** isoechoic with hypoechoic surface, and **(D)** hypoechoic with isoechoic surface. DUS, duplex ultrasound; ICA, internal carotid artery.

CEA has been accepted as a safe and effective procedure for optimized primary and secondary prevention of recurrent neurological symptoms and stroke for symptomatic or asymptomatic patients with moderate to severe carotid stenosis [17–19]. Due to the prophylactic nature of CEA, the perioperative stroke/death rate should be <6% among symptomatic patients and the risk associated with surgery is less than 3% among asymptomatic men [20]. However, apart from perioperative complications, restenosis of the operated site can be problematic during long-term follow-up. Carotid restenosis is not uncommon, and the incidence of symptomatic restenosis, usually occurring within 2 years after CEA, has been reported to range between 0.6% and 3.6% [6, 8]. In a recent meta-analysis, Fokkema et al. [21] observed that two-thirds of reinterventions for restenoses were undertaken for patients with asymptomatic lesions, suggesting that many surgeons and interventional radiologists were reluctant not to reintervene. However, considering that most restenoses are generally benign and do not require reintervention [4], controversy exists regarding whether the occurrence of restenosis after CEA is

**Table 3. Factors associated with carotid restenosis after CEA.**

|  | Univariable analysis |  | Multivariable analysis |  |
|---|---|---|---|---|
|  | HR (95% CI) | P-value | HR (95% CI) | P-value |
| Old age | 0.95 (0.90–1.01) | 0.09 | 0.95 (0.90–1.01) | 0.09 |
| Female sex | 3.01 (1.09–8.35) | 0.03 | 3.05 (1.09–8.47) | 0.03 |
| BMI | 1.10 (0.94–1.29) | 0.24 | NA | NA |
| Hypertension | 1.04 (0.34–3.14) | 0.95 | NA | NA |
| Diabetes mellitus | 1.09 (0.45–2.68) | 0.85 | NA | NA |
| Dyslipidemia | 2.00 (0.72–5.52) | 0.18 | NA | NA |
| Smoking | 0.75 (0.31–1.82) | 0.52 | NA | NA |
| CKD | 1.30 (0.43–3.90) | 0.64 | NA | NA |

BMI, body mass index; CEA, carotid endarterectomy; CKD, chronic kidney disease; CI, confidence interval; HR, hazard ratio; NA, not applicable.

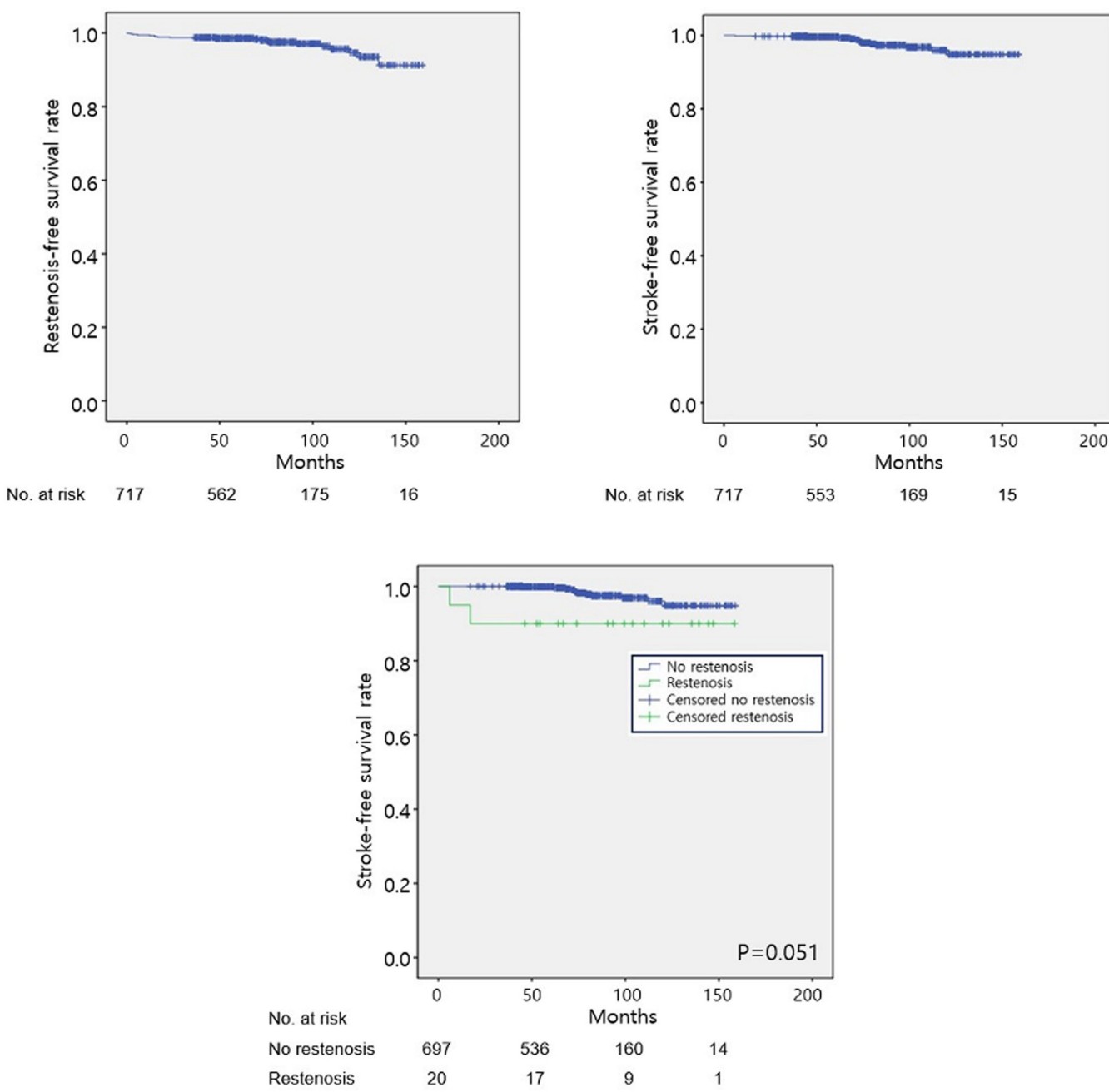

**Fig 2.** Cumulative incidence of (A) restenosis-free and (B) stroke-free survival rates of the entire study sample. (C) Stroke-free survival rate among patients with restenosis (green) and without restenosis (blue).

associated with an increased risk of neurologic events, as well as regarding which patients with asymptomatic restenosis will benefit most from reintervention [2, 8, 22, 23].

The exact pathogenesis of carotid restenosis is not fully understood [2]. Early carotid restenosis, usually developing in the first 6–12 months after CEA, is known to involve an inflammatory reaction leading to the formation of a plaque rich in fibroblasts and smooth muscle cells—a phenomenon known as myointimal hyperplasia [5, 9]. In contrast, late carotid restenosis usually develops after 24–36 months and is mainly attributed to the recurrence and progression of carotid atherosclerotic processes [24, 25]. Unlike late fibrotic changes, early "proliferative"

lesions are known to be characterized by inflammatory and proliferative processes and a tendency to cause neurologic events [2].

Carotid DUS has become an accurate noninvasive diagnostic tool for estimating the degree of postoperative restenosis, and it has been shown to be comparable to angiography [26]. After CEA, patients have been regularly followed up by carotid DUS imaging [27]. The rationale for surveillance is that disease progression on the contralateral side and restenosis on the operated-on carotid artery can reliably be detected, providing an opportunity for timely intervention to reduce the risk of neurologic events among asymptomatic patients [10]. However, the evidence for long-term follow-up using DUS is not clear, and a recent systematic review by Al Shakarchi et al. [10] reported that routine postoperative DUS surveillance after CEA is not necessary, especially if the results from early DUS are normal. They showed that the reported early rate of restenosis is low (2.8%), and of patients who had a normal early DUS scan, only 2.8% (95% CI, 0.7–6%) developed significant restenosis and 0.4% (95% CI, 0–0.9%) underwent reintervention for their restenosis during follow-up [10].

In our study, we evaluated the morphologic characteristics associated with restenosis using follow-up DUS. Most restenoses (n = 14, 70%) were observed as homogenous isoechoic lesions with regular surfaces, and 3 restenoses (15%) appeared as isoechoic lesions with hypoechoic surfaces. Among the 3 isoechoic lesions with hypoechoic surfaces, 2 were associated with neurologic events. Although 2 symptomatic restenoses were associated with specific morphologic characteristics on follow-up DUS, a causal relationship between specific DUS findings and neurologic events could not be identified. Additional large cohort studies are required.

Female sex, diabetes mellitus, and dyslipidemia are proposed as independent predictors of restenosis after CEA, and smoking is known to be associated with an increased rate of restenosis [6]. The risk of restenosis has been known to decrease over time, from 10% in the first year to 3% in the second year and only 1% per year thereafter [8]. In the present study of Asian patients who underwent CEA and were regularly followed up by carotid DUS, we found that the restenosis incidence was 2.8% and that there was a higher risk of late stroke among patients with carotid restenosis compared with those without restenosis, with a non-significant trend (P = 0.055). Most restenoses developed within 2 years after CEA (9/20, 45%), and most reinterventions were also performed within 2 postoperative years (8/11, 72.7%) and in association with asymptomatic restenosis (9/11, 81.8%). Although the actual incidence of stroke among patients with restenosis was not determined in our analysis, we presume that the incidence of restenosis-related stroke was very low. Despite the small number of events in our study sample, female sex was significantly associated with carotid restenosis after CEA; however, we could not analyze variables associated with late neurologic events.

This study had some limitations. The retrospective nature of this study made it susceptible to information biases. Because of the small sample size and small number of events, it was not feasible to analyze clinical variables associated with restenosis-related late neurologic events. Furthermore, due to the varied follow-up interval between the patients, the actual progression speed could only be assumed. Finally, as with all observational studies, we cannot draw conclusions about causality, and our results should be considered as hypothesis-generating rather than conclusive.

In conclusion, despite the potential limitations, our results suggest that carotid restenosis is commonly identified within 2 years after CEA and that there is a non-significant trend toward a higher risk of stroke among patients with carotid restenosis.

## Supporting information

**S1 File. Data of 660 consecutive patients who underwent 717 CEAs.**
(XLSX)

## Author Contributions

**Conceptualization:** Yong-Pil Cho.

**Data curation:** Hyangkyoung Kim, Eunae Byun, Min-Jae Jeong, Hee Sun Hong, Youngjin Han, Tae-Won Kwon, Yong-Pil Cho.

**Formal analysis:** Hyangkyoung Kim, Eunae Byun, Min-Jae Jeong, Hee Sun Hong, Youngjin Han, Yong-Pil Cho.

**Investigation:** Hyangkyoung Kim, Eunae Byun, Min-Jae Jeong, Hee Sun Hong, Tae-Won Kwon, Yong-Pil Cho.

**Methodology:** Hyangkyoung Kim, Eunae Byun, Hee Sun Hong, Youngjin Han, Tae-Won Kwon, Yong-Pil Cho.

**Resources:** Yong-Pil Cho.

**Supervision:** Min-Jae Jeong, Tae-Won Kwon, Yong-Pil Cho.

**Validation:** Hyangkyoung Kim, Eunae Byun, Min-Jae Jeong, Hee Sun Hong, Youngjin Han, Tae-Won Kwon, Yong-Pil Cho.

**Visualization:** Hyangkyoung Kim, Eunae Byun, Min-Jae Jeong, Hee Sun Hong, Youngjin Han, Tae-Won Kwon, Yong-Pil Cho.

**Writing – original draft:** Hyangkyoung Kim, Yong-Pil Cho.

**Writing – review & editing:** Yong-Pil Cho.

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
