## [Decision Letter · Decision Letter 0]

14 Dec 2020

Duplex ultrasound findings and clinical outcomes of carotid restenosis after carotid endarterectomy

PONE-D-20-36653

Dear Dr. Cho,

We’re pleased to inform you that your manuscript has been judged scientifically suitable for publication and will be formally accepted for publication once it meets all outstanding technical requirements.

Kind regards,

Prof. Raffaele Serra, M.D., Ph.D

Academic Editor

PLOS ONE

Additional Editor Comments (optional):

good and timely manuscript on an important issue. the article is very well written.

Reviewers' comments:

Reviewer's Responses to Questions

**Comments to the Author**

1. Is the manuscript technically sound, and do the data support the conclusions?

Reviewer #1: Yes

2. Has the statistical analysis been performed appropriately and rigorously? 

Reviewer #1: Yes

3. Have the authors made all data underlying the findings in their manuscript fully available?

Reviewer #1: Yes

4. Is the manuscript presented in an intelligible fashion and written in standard English?

Reviewer #1: Yes

5. Review Comments to the Author

Reviewer #1: well written and analyzed - topic of interest to vascular surgeons and vascular medicine - aut

authors used appropriate criteria to identify carotid restenosis - conclusions based on the study data.

The authors provided data for re-interventions for progressive stenosis - the nature of re-stenosis was provided in the images.

The Discussion was appropriate for the topic

6. PLOS authors have the option to publish the peer review history of their article (what does this mean?). If published, this will include your full peer review and any attached files.

Reviewer #1: No

---

## [Editor Report · Acceptance letter]

16 Dec 2020

PONE-D-20-36653 

Duplex ultrasound findings and clinical outcomes of carotid restenosis after carotid endarterectomy 

Dear Dr. Cho:

I'm pleased to inform you that your manuscript has been deemed suitable for publication in PLOS ONE. Congratulations! Your manuscript is now with our production department. 

Kind regards, 

on behalf of

Prof. Raffaele Serra 

Academic Editor

PLOS ONE